# Programmable quantum emitter formation in silicon

K. Jhuria ®[1] ✉, V. Ivanov[1,5], D. Polley[2,6], Y. Zhiyenbayev[2], W. Liu ®[1], A. Persaud ®[1], W. Redjem[2,7], W. Qarony[2], P. Parajuli[3], Q. Ji[1], A. J. Gonsalves[1], J. Bokor ®[2], L. Z. Tan ®[3], B. Kanté ®[2,4] & T. Schenkel ®[1] ✉

Silicon-based quantum emitters are candidates for large-scale qubit integration due to their single-photon emission properties and potential for spin-photon interfaces with long spin coherence times. Here, we demonstrate local writing and erasing of selected light-emitting defects using femtosecond laser pulses in combination with hydrogen-based defect activation and passivation at a single center level. By choosing forming gas ($N_2/H_2$) during thermal annealing of carbon-implanted silicon, we can select the formation of a series of hydrogen and carbon-related quantum emitters, including T and $C_i$ centers while passivating the more common G-centers. The $C_i$ center is a telecom S-band emitter with promising optical and spin properties that consists of a single interstitial carbon atom in the silicon lattice. Density functional theory calculations show that the $C_i$ center brightness is enhanced by several orders of magnitude in the presence of hydrogen. Fs-laser pulses locally affect the passivation or activation of quantum emitters with hydrogen for programmable formation of selected quantum emitters.

Silicon (Si)-based quantum emitters are emerging as viable candidates for quantum computing, sensing, networking, and communication owing to bright photon emission in the telecom band, scalability, and ease of integration with both electronics and photonics[1–3]. Prominent emitters have been revisited recently including the W, G, and T centers which involve common elements from standard Si processing in their structure (i.e., hydrogen (H), carbon (C)). Optimization of local (single) center formation is being pursued in process flows that include ion implantation, thermal annealing, and local excitation with focused ion beams and laser pulses[1–10]. Local laser-driven excitation has been optimized for (single) color center formation in high bandgap semiconductors such as diamond, SiC, and boron nitride[11–13], including with in situ feedback for deterministic single center formation. Laser processing of Si has enabled (local) annealing, doping, and defect engineering, e.g., for applications in electronic device formation and photovoltaics[4,5,14–16]. Here, we demonstrate programmable defect center formation with local writing and erasing of selected light-emitting defects using fs laser pulses in combination with hydrogen-based defect activation and passivation. We demonstrate this approach with G centers (a pair of two C atoms at substitutional sites paired with the same Si self-interstitial) along with $C_i$ centers (a single C atom at an interstitial site in the silicon lattice). This local processing approach for engineering Si quantum emitters with fs laser pulses paves the way toward large-scale integration of selected quantum emitters and the realization of Si-based quantum networks.

G centers have been realized using standard ion implantation followed by rapid thermal annealing at different temperatures and times in silicon-on-insulator (SOI) substrates[3,17,18]. Interestingly, the $C_i$ center, which only involves a (Si–C)$_{Si}$ split-interstitial pair has not received a lot of attention to date[19–22]. We report a simple recipe to

[1]Accelerator Technology and Applied Physics Division, Lawrence Berkeley National Laboratory, Berkeley, CA, USA. [2]Department of Electrical Engineering and Computer Sciences, University of California, Berkeley, CA, USA. [3]Molecular Foundry, Lawrence Berkeley National Laboratory, Berkeley, CA, USA. [4]Materials Sciences Division, Lawrence Berkeley National Laboratory, Berkeley, CA, USA. [5]Present address: Virginia Tech National Security Institute, Blacksburg, VA, USA. [6]Present address: Department of Physics, BITS Pilani-Hyderabad Campus, Telangana, India. [7]Present address: College of Nanoscale Science and Engineering, SUNY Albany, Albany, NY, USA. ✉e-mail: kaushalya@lbl.gov; t_schenkel@lbl.gov

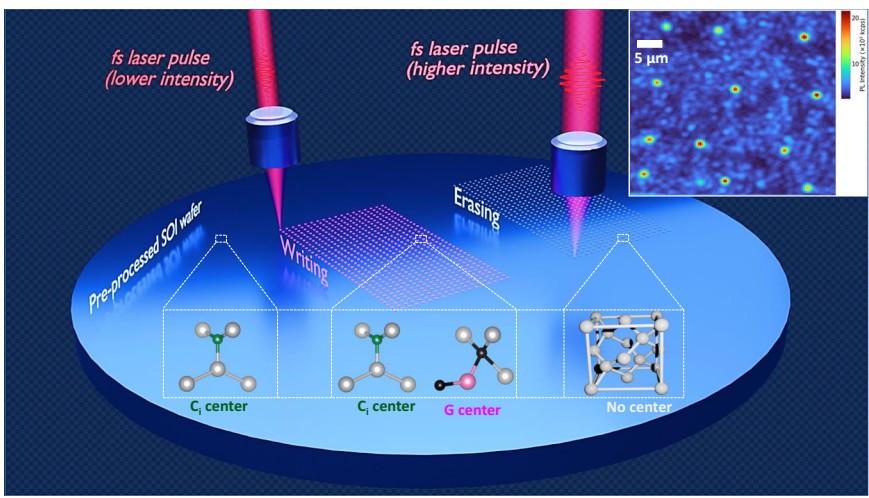

**Fig. 1 | Programmable quantum emitter formation with femtosecond (fs) laser pulses in silicon-on-insulator (SOI).** Artistic representation of the fs laser irradiation approach to locally write and erase G, and $C_i$ centers in SOI. The G center is a pair of two carbon atoms at substitutional sites (black sphere) combined with the same Si self-interstitial (pink sphere), whereas the $C_i$ consists of a pair of an interstitial carbon (green sphere) and a substitutional Si atom (gray sphere) in the Si lattice. A single fs laser pulse (pulse duration of 90 fs), with wavelength centered at 800 nm, was used for irradiation at varied fluences to locally form and erase quantum emitters. Three different areas are highlighted to represent the workflow starting with a pre-processed SOI wafer with ensemble $C_i$ centers formed after ion implantation and rapid thermal annealing under forming ambiance (area on the left). The second area (in the middle) represents the writing of G centers along with modified $C_i$ centers on the pre-processed SOI sample via single fs laser pulse irradiation at relatively low fluences (<30 mJ/cm²). The third area (on the right) shows the erasing of quantum emitters after irradiation with a relatively higher laser fluence (still much below the melting threshold for Si). A photoluminescence hyperspectral scan with fs laser pulse irradiation spots processed at a fluence of 12 mJ/cm² is shown in the top-right corner.

form $C_i$ centers, as well as T-centers[2,23], using standard ion implantation and rapid thermal annealing. $C_i$ and T-centers form and G-centers are passivated when we replace inert gases (i.e., Ar and $N_2$) with forming gas ($H_2$: 10%, $N_2$: 90%) (see "Methods" and Supplementary Note 7, and Supplementary Figs. 11–13 for details on the dependence of forming gas annealing parameters on the selective formation of quantum emitters) during the annealing process. We can then further program and select the presence of specific centers, e.g., $C_i$ or G-centers, by fs laser-based writing and erasing. Electron paramagnetic resonance (EPR) measurements by Watkins and Brower[21], revealed two optically addressable spin ½ charge states, which qualify the $C_i$ center as a potential spin-photon interface candidate. While the T-center has been widely studied and has been qualified as a leading spin-photon qubit candidate, less is known about the formation and level structure of $C_i$ centers. Here, we focus on $C_i$ centers and demonstrate programmable single-center formation based on a combination of forming gas annealing and fs laser processing conditions in a family of carbon and hydrogen-related quantum emitters in silicon.

## Results

### Formation, writing, and erasing of light emitting centers in SOI with fs laser pulses

An artistic view of writing and erasing of quantum emitters in SOI using fs laser pulses is shown in Fig. 1 along with a structural representation of G and $C_i$ centers. Programmable writing and erasing of color centers can be achieved by changing the fs laser fluence. A photoluminescence (PL) hyperspectral image presented in the top right corner of Fig. 1 shows twelve different isolated G and $C_i$ centers after irradiation with single fs laser pulse on each spot at 12 mJ/cm² fluence. The PL map was taken with a 1250 nm long pass filter placed before the superconducting nanowire single photon detector (SNSPD) to attenuate the background signal. Figure 2a shows PL spectra at each step in the process flow starting with an as-received SOI after carbon ion implantation and forming gas annealing (pre-processed SOI) resulting in bright $C_i$ center formation. The presence of hydrogen during annealing plays a pivotal role in the formation of $C_i$ centers while passivating the G centers. This is in contrast to many earlier reports,

where G-centers dominate emission spectra after carbon ion implantation and thermal annealing in an inert gas ambient[18,22]. Telecom band wavelengths (i.e., 1200–1600 nm) were scanned to detect the resulting color centers, the selected range is shown for better visualization. Generally, it has been observed that the linewidth of quantum emitters broadens when transitioning from bulk Si to SOI due to the Si/$SiO_2$ interface-induced stress and strain[24]. Interestingly, the $C_i$ centers formed with the above-mentioned recipe provide narrow linewidth of ~0.03 nm (4.2 GHz), a measurement value that is limited by the spectrometer resolution (see Supplementary Note 3 and Supplementary Fig. 3 for a high-resolution spectrum). This can be attributed to the diffusion of H into Si during the annealing process and the formation of H clusters and trapping of impurities to compensate for strain at the Si/$SiO_2$ interfaces[25]. We have further computed the linewidth broadening of the $C_i$ center as a function of strain and stress and compare it with the G center (see Supplementary Note 5.3 and Supplementary Fig. 7 for details). A PL saturation curve was also recorded for $C_i$ centers along with the temperature response to check its stability and robustness (see Supplementary Note 2 and Supplementary Fig. 2).

SOI samples pre-processed by carbon ion implantation and thermal annealing in forming gas were then subjected to fs laser pulses of varying fluences in the range ~16–48 mJ/cm². This fluence range is ~4 times lower than the damage threshold of Si for fs-laser pulses[26,27]. Hence the Si lattice is not damaged, but fs-pulses in this fluence range can act on the H-bonds to atoms and defect structures in the silicon matrix. Hence, fs-laser pulses enable us to write and erase quantum emitters locally. The last three curves in Fig. 2 show the PL spectra obtained after fs-laser irradiation at different laser fluences demonstrating first the writing of G centers along with modifying the density of pre-existing $C_i$ centers followed by partial and complete erasure of G and $C_i$ centers, respectively, at slightly higher laser fluences. The final curve presents the reoccurrence of G and $C_i$ centers after irradiation with a higher laser fluence pulse of ~44.5 mJ/cm². The PL emission peak intensity corresponding to G and $C_i$ centers is shown in Fig. 2b before and after fs irradiation indicating the writing and erasing of these emitters (see Supplementary Note 4 and Supplementary Fig. 4 for spectra corresponding to the intermediate laser fluences). While the

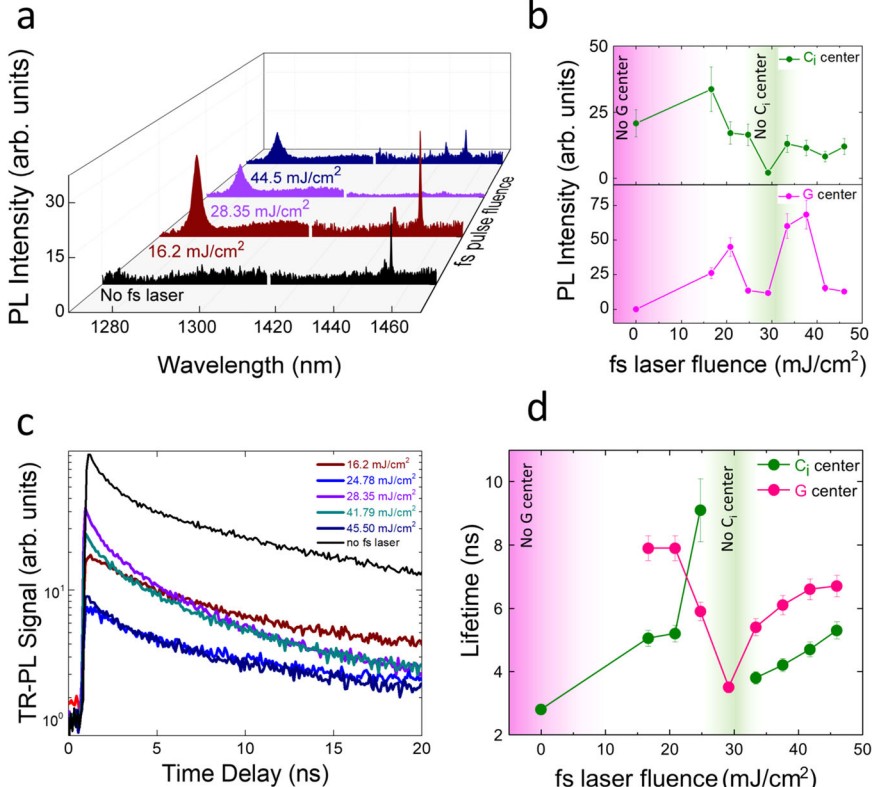

**Fig. 2 | Writing and erasing of G and $C_i$ center with fs laser pulses below the damage threshold of Si. a** Process flow represented by measuring the PL spectra starting from the pre-processed SOI sample (emission ~1452 nm, corresponding to the $C_i$ center after carbon ion implantation (7e13 C/cm$^2$ fluence) followed by rapid thermal annealing at 800 °C for 120 s under forming gas ambiance). The last three curves show the PL spectra obtained after fs irradiation at different laser fluences in order to first write G centers along with modifying the density of pre-existing $C_i$ centers (~16 mJ/cm$^2$), followed by partial and complete passivation of G and $C_i$ centers, respectively, at higher fluences (~30 mJ/cm$^2$). The final curve presents the reoccurrence of G and $C_i$ centers after irradiation with an even higher fluence pulse (i.e., 44.5 mJ/cm$^2$). The damage threshold in our experiments was >100 mJ/cm$^2$. **b** PL emission peak intensity of G and $C_i$ centers after irradiation with fs pulses of varying energies. **c** TR-PL signal from the $C_i$ centers before and after the fs laser irradiation to extract optical lifetimes. **d** Optical lifetimes as a function of fs laser fluence for both G and $C_i$ centers extracted by fitting the TR-PL signal with a first-order decay function. Error bars shown in (**b**–**d**) are statistical errors from sample to sample that map to variations in the pulse energy of the fs laser output observed in our experiments. Pink and green color-band in (**b**) and (**d**) are representative of the likelihood of the absence of G and $C_i$ centers, respectively.

---

in-depth understanding of the mechanism governing the writing and erasing of $C_i$ centers with single pulse fs laser irradiation requires further exploration, it can be essentially understood with single photon absorption leading to energy transfer to the electronic system, which can be further transferred to the silicon lattice, that in turn couples to the H atoms bonded to carbon and silicon atoms in defect complexes. For the range of laser fluences shown in Fig. 2, 16–48 mJ/cm$^2$, we find that the Keldysh parameter varies from ~19 to 6[28]. This places the Keldysh parameter in the photoionization regime, as it is above the threshold value of 1.5. In our samples, photoionization likely takes place via single photon absorption, as the excitation energy of our lasers (1.55 eV) is above the band gap of Si (1.12 eV). Hot carriers generated by photoionization lose energy to the Si lattice via electron–phonon coupling. We expect that the excitation of the vibrational modes of Si–H bonds and C–H will be linked to the formation or erasing of the $C_i + H$ centers. C–H stretching vibrational frequencies are typically around 0.36 eV[29], while Si–H stretching frequencies are typically around 0.26 eV[30]. For hot carriers at 0.43 eV above the band edge, there is enough energy for the excitation of these vibrational modes to start the process of H migration to or from a $C_i$ center.

Fs-laser fluences well below the Si damage threshold enable defect center reconstruction into optically dark states and the removal of hydrogen from $C_i$ centers. Increasing the laser fluence to ~40 mJ/cm$^2$ can aid the redistribution of H and leads to the re-

activation of $C_i$ centers[31]. Writing of G centers can be understood in a similar way where the breaking of a H-bond in the vicinity of an optically inactive G center (A-configuration) may lead to the formation of an optically active G-center in the B configuration[32] (see discussion section for more details). Time-resolved photoluminescence (TR-PL) measurements were also performed before and after the fs irradiation to extract the non-radiative lifetimes for both G and $C_i$ centers. The lifetime of $C_i$ centers was found to be ~3 ns before the fs laser irradiation and changed between 3 and 8 ns for different laser fluences. A similar lifetime range was also found for the G centers after fs irradiation. It can be noted that a slight change in the fs laser fluence can change the lifetime of the quantum emitters and hence provides fine control over the quality of the emitters[33].

## Isolated G and $C_i$ center formation post fs laser irradiation

In order to find the right laser fluence to achieve single center level control, we have scanned a wide range of laser fluences, from well below the melting threshold of silicon (~10 s of mJ/cm$^2$), to well above the damage threshold of silicon (over 300 mJ/cm$^2$). We observe that for the fluence range of ~15–50 mJ/cm$^2$, centers are formed and erased at the ensemble level (as shown in Fig. 2). For laser fluences well above the melting threshold (i.e., ~1000–6000 mJ/cm$^2$), no $C_i$ centers were observed and the PL is dominated by G and W centers. (see Supplementary Note 8 and Supplementary Fig. 14 for details).

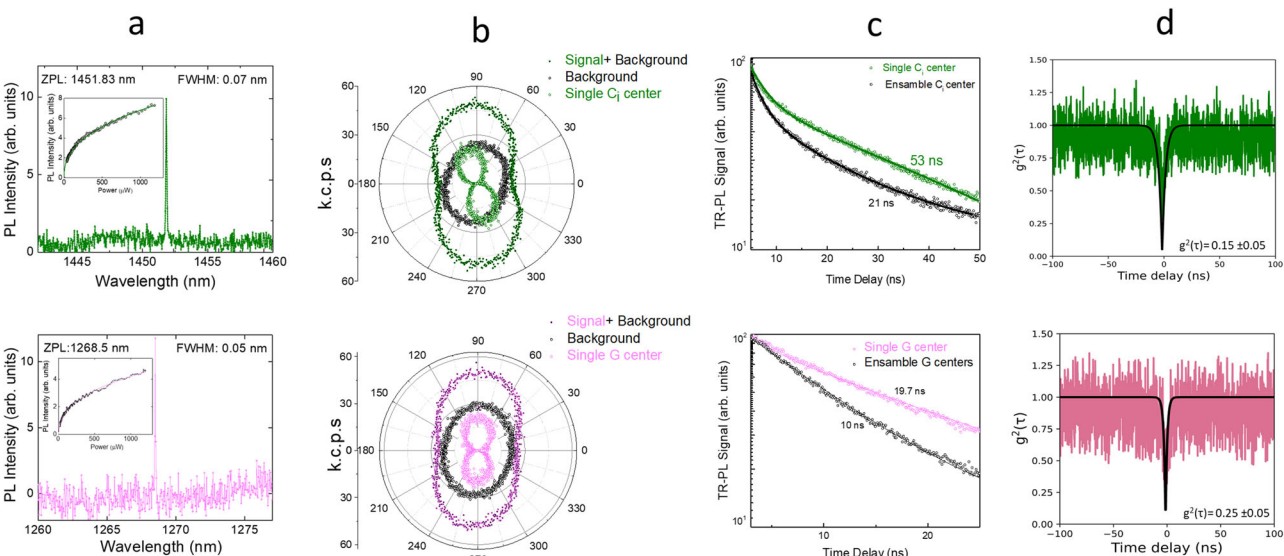

**Fig. 3 | Isolated of $C_i$ and G center formation post fs pulse irradiation. a** PL spectrum (recorded with 600 g/mm grating with a resolution of 0.05 nm) corresponding to the single $C_i$ (top row) and G (bottom row) centers along with a typical green laser excitation power-dependent PL (insets). **b** Polarization sensitivity of the PL emission from $C_i$ and G center (top and bottom row, respectively) along with ensemble background mapping to the single center emission. **c** Optical lifetime measurement for isolated $C_i$ and G center (top and bottom row, respectively) along ensemble background. (Significantly longer optical lifetimes were observed for the isolated G and $C_i$ centers in comparison to their ensemble counterpart). **d** Background corrected second-order auto-correlation signal with phenomenological fit for both $C_i$ and G centers (top and bottom row, respectively) using Hanbury-Brown and Twiss (HBT) interferometer. The experimental value of the function at zero time delay was observed to be much below 0.5, proving the single photon emission from both centers. All the measurements on an irradiation spot with fs pulse fluence of 8 mJ/cm$^2$.

Single, isolated G and $C_i$ quantum emitters were formed at very low laser fluences of ~8–12 mJ/cm$^2$. Figure 3a shows the PL spectrum (recorded with 600 g/mm grating with a resolution of 0.05 nm) corresponding to the single $C_i$ (top row) and G (bottom row) centers along with a typical green laser excitation power-dependent PL (insets). Figure 3b shows the polarization sensitivity of PL from both centers at single and ensemble background. Ensemble background signal was probed a micron away from the single center spot. TR-PL signal corresponding to the background ensemble and the signal $C_i$ and G centers along with extracted optical lifetimes is shown in Fig. 3c. Polarization-sensitive PL spectra, a significantly longer lifetime for the single center in comparison to the ensemble background are some initial indicators for the presence of isolated single centers. Second-order auto-correlation measurement for both $C_i$ and G centers using a Hanbury-Brown and Twiss (HBT) interferometer was then performed at 160 μW green laser excitation power to test for single photon emission in selected areas. A background-corrected experimental and phenomenological fit for both emitters is shown in Fig. 3d (see Supplementary Note 6.2, Supplementary Fig. 9, Supplementary Eq. (1) for details). The value of the function at zero time delay $g^2(\tau = 0)$ was observed to be well below 0.5, demonstrating the single photon emission from both centers. Measurements shown in Fig. 3 were performed following local programming of center populations with a fs laser pulse intensity of only 8 mJ/cm$^2$. A systematic study on several such isolated $C_i$ and G centers after fs pulse irradiation (in the fluence range of 8–12 mJ/cm$^2$) has been done by monitoring their corresponding PL spectra, optical lifetimes, polarization sensitivity, long-time stable emission under excitation to the PL (see Supplementary Notes 6.1, 6.3 and Supplementary Figs. 8 and 10 for details). As discussed in the first section, for these low laser fluences, the formation of single photon emitters in our SOI samples is likely mediated by single photon absorption in the photoionization regime, and similar to fs laser-based single emitter formation in diamond and SiC[13,34].

## First-principles calculations of $C_i$ centers

We performed first-principles calculations of $C_i$ centers, using a computational workflow described in earlier publications[35], with atomic defect structure corresponding to a split $(Si-C)_{Si}$ pair sharing a substitutional site (Fig. 4a)[19] (see "Methods"). The energy level diagram for the neutral charge state of the $C_i$ center is shown in Fig. 4e, while the diagrams for other charge states and a plot of the chemical potential-dependent formation energies can be found in Supplementary Note 5.1 and Supplementary Fig. 5. We find that the 0, −1, −2, and −3 charge states of the $C_i$ center are stable within the Si gap, with the −1 charge state being stable in intrinsic Si. This is in contrast to prior work[19], which found the +2, +1, and 0 charge states to be stable, albeit with a different correction scheme and a smaller supercell size, which leads to much larger errors due to periodic image charges. As the −3 charge state is only stable in heavily n-doped Si, and the −2 charge state has all defect levels occupied, we focus on the −1 and 0 charge states, as well as the +1 charge state due to it being found stable in prior work.

The computed zero-phonon lines and transition dipole moments of the −1, 0, and +1 states are shown in Table 1. As the number of electrons in the center changes, the energies of newly occupied defect levels shift, but the localized defect wavefunctions remain relatively unchanged, localizing to the carbon atom for the lower defect levels, and to the Si for the upper defect levels. The optical properties of these defects can be considered in several ways. First, the Kohn-Sham energy differences (ΔKS) are 968 meV, 817 meV, and 812 meV for the −1, 0, and +1 charge states, respectively, and are known to be a decent estimate for the zero-phonon line (ZPL) of defects, deviated by ~100–150 meV[35,36]. The computed ZPLs for the −1, 0, and +1 charge states using the constrained occupation method are 571 meV, 569 meV, 568 meV, respectively. Due to the finite size effects of the supercell and a limited number of k-points, these values can also deviate from the true ZPL by several tenths of an eV[32,35]. Given these considerations, the different charge states of this split $(Si-C)_{Si}$ pair structure may explain the multiple experimentally observed ZPL

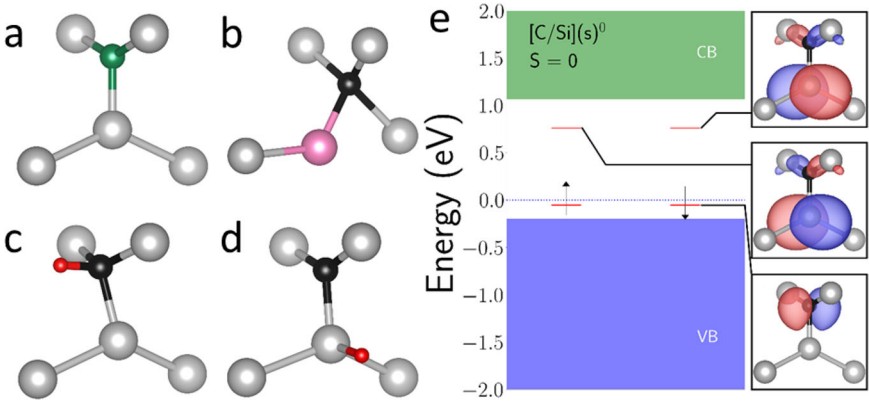

**Fig. 4 | Defect levels, structures, and modifications of the $C_i$ center.** Structure of **a** the $C_i$ center, **b** the displaced "B" configuration of the $C_i$ center, and modified versions of the $C_i$ center with H bonded to either **c** the carbon, or **d** the Si atom. Atoms are colored as follows—Si (gray), carbon (black, green), H (red), Si self- interstitial (pink). **e** Energy level diagrams for the neutral charge state of the $C_i$ center, with conduction band (green), valence band (blue), localized defect levels (red), and electron occupation (black arrows) indicated. Panels on the right show the real space wavefunctions corresponding to each localized defect level.

**Table 1 | Energies and dipole moments of $C_i$ center charge states and modifications**

| Defect | Zero-phonon line (ZPL) (meV) | Spin channel | Squared transition dipole moment (TDM) (Debye$^2$) | Relative shift (meV) |
|---|---|---|---|---|
| $C_i$ center (−1) | 571 | Down | $0.169 \times 10^{-5}$ | +2 |
| $C_i$ center (0) | 569 | N/A | $0.337 \times 10^{-5}$ | |
| $C_i$ center (+1) | 568 | Down | $0.482 \times 10^{-8}$ | −1 |
| $C_i$ center "B" (0) | 655 | N/A | 0.118 | +86 |
| $C_i$ + H Type 1 (0) | 103$^2$ | Down | $0.29 \times 10^{-2}$ | +463 |
| | 528 | Up | 0.483 | −41 |
| $C_i$ + H Type 1 (+1) | 611 | N/A | 0.632 | +42 |
| $C_i$ + H Type 2 (0) | 847 | Down | $0.28 \times 10^{-2}$ | +278 |
| | 559 | Up | 0.721 | −10 |
| $C_i$ + H Type 2 (+1) | 581 | N/A | 0.633 | +12 |
| $C_i$ + H Type 3 (0) | 1174 | Down | 2.52 | +578 |
| | 592 | Up | $0.902 \times 10^{-4}$ | +23 |
| $C_i$ + H Type 3 (+1) | 591 | N/A | 2.20 | +22 |

Computed ZPL and squared transition dipole moment (TDM) for different charge states and structural modifications of the $C_i$ center in Si. For defects with a spin degree of freedom, the spin channel for the transition is indicated. The final column gives the ZPL deviation for each defect relative to the neutral $C_i$ center.

peaks. However, it should be noted that the described transitions have vanishingly small transition dipole moments, suggesting that this structure might be optically dark.

The small transition dipole moment of the $C_i$ center is the result of the two mirror plane symmetries of the defect and the localization of defect states to different atoms in the defect. This is analogous to the G center, where the A configuration with an interstitial carbon atom is relatively optically dark[32]. By distorting the $C_i$ center so that the carbon occupies the substitutional position, which we term the "B" config- uration in analogy to the optically bright G center B, the brightness of the $C_i$ center is enhanced considerably (Table 1), though this structure is 0.66 eV higher in energy. If the mirror symmetries are broken by displacing atoms of the $C_i$ center out of the plane, they are restored during the force relaxation step, suggesting this is not a stable configuration.

Exposure to forming gas during thermal annealing introduced hydrogen into the silicon lattice, leading to the possibility of generating H-related centers. Inspired by the T center, where two carbon atoms occupying a substitutional site are bound to an H atom[37], we consider modified $C_i$ center defects, where a hydrogen atom is either bound to the carbon atom in a planar configuration ($C_i$+H Type 1), bound to the carbon with an additional out of plane distortion ($C_i$+H Type 2, Fig. 4c), or bound to the Si atom ($C_i$+H Type 3, Fig. 4d). These modified defects

have slightly shifted ZPLs, and are all significantly optically brighter than the bare $C_i$ center (Table 1). They further have a spin-1/2 degree of freedom in the neutral charge state, like the T center.

Aside from the known ZPL peak at 1448 nm (856 meV), a number of other peaks at 1415.4 nm, 1441.7 nm, 1444.3 nm, 1450.8 nm, and 1453.6 nm can be seen in the experimental PL spectra, which corre- spond to respective shifts of +20 meV, +4 meV, +2 meV, −2 meV, and −3 meV in energy (see Supplementary Note 3 and Supplementary Fig. 3). Our first-principles calculations show that there are shifts in the ZPL of the same magnitude arising from charge states, structural changes, and the presence of H (last column, Table 1). Considering that the deviation of the computed 569 meV ZPL from the observed 856 meV ZPL for the lone $C_i$ center due to systematic errors of the finite unit cell size and constrained occupation method is not strongly structure dependent, we infer that variations in the charge and struc- ture of the $C_i$ center in the presence of hydrogen can explain the variety of observed bright ZPL peaks.

## Discussion
### Formation mechanism of G and $C_i$ centers
G, T, H, $C_i$ centers are part of a family of carbon-related quantum emitters in silicon[38]. The G center in Si is among the most common defect centers and can be formed readily by C ion implantation

followed by rapid thermal annealing under inert gas ambiance, as well as by proton irradiation, and laser-ion doping[39–41]. The formation of the G center follows a widely accepted two-step process. Firstly, radiation damage leads to the creation of Si self-interstitial and an accompanying $C_i$ due to ion implantation. The $C_i$, known for its high mobility, can freely migrate within the lattice until it becomes trapped by a substitutional carbon atom, $C_s$. This trapping event results in the formation of the optically inactive A configuration of the G center. However, this configuration can overcome a small potential barrier (~0.14 eV) and undergo a structural transformation to become the optically active B configuration. The B configuration comprises two C atoms located at substitutional sites, both bonded to the same Si self-interstitial[22,32].

### Passivation of G centers by forming gas annealing

Hydrogen is widely used for defect passivation in semiconductor processing, e.g., to increase minority carrier lifetimes and to reduce interface charge densities[42]. Hydrogen has been previously used to passivate many other defect centers in Si[43], and this is further demonstrated here in the effect of H on the G center. In the optically active B form of the G center, the unpaired electrons are localized to the self-interstitial Si atom and can trap H atoms. We consider two configurations for the H, in-plane, and out-of-plane (see Supplementary Note 5.2 and Supplementary Fig. 6), computing the dipole moments for transition associated with the bright G center luminescence, from the valence band maximum to the midgap localized defect level. The squared transition dipole moments (TDM) for these transitions are 0.067 Debye$^2$ and 0.0845 Debye$^2$, significantly less than the ~5 Debye$^2$ of the native G center, which can explain why G centers are not observed in the PL data from SOI samples that had been pre-processed with thermal annealing in a forming gas ambient shown above (1st curve from Fig. 2a). Complementary to the G center, the TDMs of the Ci center with additional hydrogen increase by several orders of magnitude (Table 1).

### Writing and erasing mechanism of quantum emitters in silicon under fs irradiation

While the writing and erasing of G and $C_i$ centers with fs laser in our pre-processed SOI samples could be understood as the relatively low-intensity fs-laser pulses acting mostly on H-bonds, more complicated physical mechanisms are also possible, where the Si–Si and Si–C pair get disassociated due to energy deposition and electronic excitation effects with direct fs laser irradiation, leading to a rearrangement of impurities and dopant atoms in the silicon lattice. For laser fluences near or above the melting threshold of Si, tunneling is likely the governing mechanism leading to formation of ensembles of color centers in laser spots. We suggest that this regime of laser processing near or above the melting threshold of Si is not favorable for the reliable formation of single, isolated quantum emitters in Si (see Supplementary Notes 8 and 9 and Supplementary Figs. 14 and 15 for further details on the formation of G centers on as-received SOI below damage threshold). Fs-laser pulses of varying intensity can steer these re-configuration processes and enable the programming, or selective writing and erasing of specific quantum emitters with properties optimized for selected applications, from quantum communications to quantum sensing. As a topic for future research, an in-depth understanding of these mechanisms will require simulations of non-adiabatic dynamics using methods such as time-dependent density functional theory, which has shown promise in the study of defect formation in the excited state[44–47]. We show how the choice of thermal annealing conditions with forming gas together with local fs laser pulse intensities allows for the selectively erasing or formation of specific quantum emitters. This approach for selective and programmable center formation can be used to form deterministic single center arrays or structures when in situ PL feedback is provided, as has been shown for deterministic single NV center programming in diamond[12].

In this article, we address challenges in the development of single photon emitters and spin-photon interfaces, i.e., selective and programmable local formation of qubit candidates with tailored properties. By combining standard semiconductor fabrication methods such as ion implantation and rapid thermal annealing (in forming gas) with local fs laser processing[48], we demonstrate the selective writing and, erasing of quantum emitters in SOI for a series of carbon-related quantum emitters. While the approach can be generalized to other emitters in Si, we have demonstrated it on the most common and widely studied light emitting defects in silicon, the W, G, H, and T centers, together with the re-discovered $C_i$ center in SOI wafers. The optical properties of the $C_i$ center qualify it as a highly promising candidate for applications as a single photon source and as a potential spin-photon interface due to its relatively simple structure, bright emission in the telecom S-band (comparable to the brightness of G and T centers), relatively narrow spectral linewidth, robustness, and spin degree of freedom in its ground state. The level structure of the H decorated $C_i$ center has yet to be mapped out to validate this promise. Density function theory calculations highlight the role of hydrogen in boosting the brightness of $C_i$ centers with hydrogen. This approach for selective programming, writing, and erasing of desired quantum emitters paves the way towards quantum emitter integration towards the realization of scalable quantum networks and engineering of qubits by design.

## Methods

### Ion implantation and rapid thermal annealing

Samples cut from commercial SOI wafers (220 nm thick device layer, 10–20 Ohm cm, p-type, 2 μm SiO$_2$ box) were used for this study. For the pre-processed SOI, the as-received SOI was first implanted with 38 keV carbon ions ($^{13}$C) targeting a mean implantation depth of ~115 nm (i.e., center of the 220 nm Si device layer). $^{13}$C implanted SOI was treated by rapid thermal annealing in forming gas (10% H$_2$, 90%N$_2$) at 800 °C for 120 s. RTA in forming gas leads to the repair of the implant damage, passivation of G-centers, and the formation of bright $C_i$ centers. The $^{13}$C isotope of carbon was chosen to enable differentiation of the ion-implanted carbon from any $^{12}$C that was present in the starting material (e.g., using secondary ion mass spectrometry, SIMS). $^{13}$C also enables the exploration of quantum memory in the nuclear spin state[49]. (see Supplementary Note 1 and Supplementary Fig. 1 for further details on ion implantation and rapid thermal annealing along with depth-resolved SIMS profile of common elements in SOI before and after ion implantation followed by forming gas thermal annealing).

### Fs laser irradiation

An amplified Ti:sapphire laser, working at a repetition rate of 250 kHz, with a wavelength centered around 800 nm and a pulse length of 90 fs (full width at half maxima, FWHM) was used for the experiment in single-shot mode. The pulse duration was measured via auto-correlation with an APE Pulse link 150. We used a laser spot size of $20 \times 20$ μm$^2$ for local defect center processing. The spot size was measured directly with a beam profilometer at sample position[50]. Samples were irradiated using single fs laser pulses with varied energy per pulse ranging from ~65–195 nJ (corresponding to a laser fluence range of ~16–48 mJ/cm$^2$) to form and passivate G and the $C_i$ quantum emitters. For the formation of W centers, we increased the laser fluence up to ~300 mJ/cm$^2$.

### Characterization

Photoluminescence (PL) and time-resolved photoluminescence (TR-PL) measurements were performed at 6 K using a scanning confocal

microscope for near-infrared spectroscopy. A 532 nm continuous wave and pulsed laser focused onto the sample via a high numerical-aperture microscope objective (NA = 0.85) was used for the optical excitation. The same objective was directed to a spectrometer coupled to an InGaAs camera (900–1620 nm at −80 °C). The telecom band wavelengths ranging from 1200–1600 nm were scanned with a grating of 150 g/mm in front of the camera. The absorption depth of 1 μm in Si with a 532 nm laser sets the probing depth for the quantum emitters.

## First-principles DFT calculations

In our calculations, a given defect was embedded into a $3 \times 3 \times 3$ supercell of Si, and VASP[51,52] was used to completely relax the atomic positions and obtain the electronic structure. For the electronic structure, the HSE06[53] functional was used, with the following parameters: a 450 eV energy cutoff, convergence tolerance of $10^{-10}$ eV, and a force tolerance of 0.001 eV/Å. Excited state energies were computed using the constrained occupation method[54] with the same parameters, while post-processing and real space wavefunctions were extracted using VASPKIT[55]. Formation energies were extracted from calculations using PBE functionals[56], and corrected for finite size effects using the Spinney package[57,58]. All calculations were performed at the $\Gamma$-point.

## Data availability

The data supporting the plots within this article are available from the corresponding author upon request.

## Code availability

The codes supporting the plots within this article are available from the corresponding author upon request.

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

## Acknowledgements

This work was supported by the Office of Science, Office of Fusion Energy Sciences, of the U.S. Department of Energy, under Contract No. DE-AC02-05CH11231. L.Z.T. and V.I. were supported by the Molecular Foundry, a DOE Office of Science User Facility supported by the Office of Science of the U.S. Department of Energy under Contract No. DE-AC02-05CH11231. This research used resources of the National Energy Research Scientific Computing Center, a DOE Office of Science User Facility supported by the Office of Science of the U.S. Department of Energy under Contract No. DE-AC02-05CH11231 using NERSC award NERSC DDR-ERCAP0025754. B.K. acknowledges support from the NSF QLCI programme through grant number OMA-2016245, and the NSF QuIC-TAQS award 2137645.

## Author contributions

T.S. and K.J. conceived and designed the experiments. K.J. and D.P. performed the single shot fs irradiation experiments with guidance from T.S. and J.B. K.J. performed the PL experiments with help from Y.Z. and W.R., with guidance from T.S. and B.K. K.J., V.I., and T.S. analyzed the results with inputs from D.P., Y.Z., W.L., A.P., W.R., W.Q., P.P., Q.J., A.J.G., J.B., L.Z.T., and B.K. V.I. and P.P. performed the first principles-based calculations and summarized the theoretical results in the manuscript with guidance from L.Z.T. K.J. and T.S. wrote the manuscript with inputs from V.I., D.P., Y.Z., W.L., A.P., W.R., W.Q., P.P., Q.J., A.J.G., J.B., L.Z.T., and B.K.

## Competing interests

The authors declare no competing interests.
