## [Peer Review File · Nature Communications]

REVIEWER COMMENTS

Reviewer #1 (Remarks to the Author):

This work reports local writing and erasing of silicon emitters in the telecom spectral windows using fs laser pulses. The authors demonstrate the controllable formation of G and Ci color centers depending on the forming gas annealing during ion implantation and laser pulse fluence. The authors also explain the bright emission of the Ci centers due to the presence of hydrogen. The approach demonstrated in this manuscript could be potentially used for large-scale intration of quantum emitters in silicon. I have the following comments and questions to the manuscript.

1. Though the authors propose to use their approach for scalable quantum networks, single photon emission is not demonstrated in the manuscript. The Ci centers in Fig. 2a are observed in the pre-processed SOI without fs laser. With fs laser, the Ci centers are erased, but there no strategy described, how can it be used for the controllable creation of single Ci centers.
2. The authors show that the fs irradiation with laser pulses of an implanted and annealed SOI sample leads to the formation of G centers. But single photon emission is not demonstrated. The authors should describe the strategy how single G centers can be isolated using their approach. In addition to that, the spectral width of the zero-phonon line of the G centers is larger than that of the G centers created by well-established methods. In the similar approach [arXiv:2304.03551], the spectral quality seems to be better. Can the authors comment on this?
3. I find many self-citations of the authors, but no citations of some earlier achievements of other groups that seem to be relevant to their work
 - Opt. Express 28, 26111 (2020): isolation of single G centers in silicon
 - New J. Phys. 23, 103008 (2021): co-implantation of carbon and hydrogen for the formation of T center in silicon
 - Nat. Commun. 14, 2380 (2023): spectrally programmable quantum emitters in silicon
 - Nat. Commun. 14, 361 (2023): theory of the formation of carbon-related color centers in silicon
4. The use of words “quantum emitter” in the title and in the main text regarding the experimental results is misleading. The Ci centers emit photons as any ensemble of color centers, but where is “quantum emission” there? In addition to that, “programmable” means that the Ci and G centers

center can be activated and erased on demand, independent on the previous state. Particularly, the reactivation after erasing should be demonstrated and the number of such cycles should be discussed. Otherwise, the use of word “programmable” in the title is also misleading. Perhaps, “reconfigurable” or similar would be better.

5. The authors explain the high brightness of the Ci center due to the distortion by the hydrogen atom. The transition dipole moment is calculated for different configurations. To understand the feasibility of the isolation of single Ci centers, it is necessary to compare it with other centers, for instance G and/or T center.

6. If the presence of hydrogen for high brightness of the Ci centers is important, the dependence on the hydrogen concentration should be provided or at least discussed. Is there a difference if hydrogen is implanted instead of using it as forming gas during the annealing process? What is the dependence on the H₂/N₂ ratio in the forming gas?

Summarizing, the authors present a potentially interesting approach for the fabrication of telecom emitters in silicon and control their brightness. On the other hand, the “programmable” and “quantum emitter” are not really demonstrated in the manuscript and the authors oversell their results. Though I think that all minor issues can be resolved in the revised version, the experimental verification of quantum emission requires new experiments. Because this missing piece is the most important part, I do not recommend publication in Nature Communications in the present form.

Reviewer #2 (Remarks to the Author):

In the manuscripts, the authors demonstrated two main results in the experiments:

1. Local writing and erasing of G centers and Ci centers using fs laser pulses in pre-processed SOI sample
2. Writing and erasing of G centers with direct fs laser pulses in as-received SOI

The main experimental data is the PL spectra from emitters induced by different laser fluences. I think there are some loose arguments in the manuscript. Please find comments below:

1.The title of the manuscript is “Programmable quantum emitter formation in silicon”. However, there is no evidence that the defects induced by fs pulse have quantum nature. To prove this claim, HBT experiments are necessary to verify whether the defects possess antibunching properties. Therefore, at least at current stage, the author's discussion in terms of quantum emitters is inappropriate. In addition, the authors do not fully demonstrate the programmable features. The authors need to explicitly give the repeatability, spatial positioning accuracy, wavelength stability of the color centers prepared by this method. At the same time, I suggest that the authors give the intensity uniformity of the arrays of color centers by fluorescence mapping (a better demonstration than Fig.1).

2.The Ref 4 reported the creation of W and G-centers by femtosecond laser annealing for the first time on SOI substrates, taking both C-doped and pristine SOI wafers. They can selectively erase the G centers while improving the quality the W centers' emission. The author should be more explicit about the innovations and improvements of this manuscript compared to the previous works.

3.The most significant data in the manuscript is the variation of the spectra of laser-induced sites with the laser power. As for the fabrication method of quantum emitters in silicon, more optical properties characterization of quantum emitters are needed for a more comprehensive statement of the laser-created color centers in silicon. For example, the long-term stability of the color centers under the light excitation. The current measurements made are too basic to substantiate the claims of significance.

4.Line 97-192 in the main text: Considering the bandgap of silicon, the main absorption mechanism of femtosecond lasers at 800 nm should not be multiphoton absorption. At the same time, the discussion of the physical mechanisms in this section is too general to be convincing. The authors should give a specific theoretical analysis and simulation of the interaction mechanism in order to relate it to the high-quality center generation. The same problem occurs in the discussion of linewidths, where the authors do not give a convincing reason for the physical origin of such a narrow linewidth.

5.Does the transition from single color center to multiple color centers occur as the laser energy increases?

Reviewer #3 (Remarks to the Author):

Jhuria et al. report on generating interstitial carbon centers (Ci) and G-centers in silicon using femtosecond (fs) pulsed lasers. They also report on a hydrogen-based annealing recipe that enhances the Ci centers and suppresses the G-centers. Additionally, they present density functional theory calculations that analyze the Ci center in the presence of hydrogen.

Significance:

As noted by the authors, pulsed laser annealing for writing color centers has been widely studied in many wide-bandgap materials already (for example, see Castelletto et al, *Nanomaterials* (Basel). 2021 Jan; 11(1): 72). Nevertheless, examining the role of fs laser irradiation in silicon specifically and the possible centers available for writing is a valuable exercise as silicon-based color centers occupy an increasingly prominent role in the color center community.

1- It would be useful to have a short comparison of figures of merit to the author's previous published work generating G-centers using pulsed ions – i.e., how do the centers compare? Additional context for the quality of the color centers created using this technique compared to other methods (non-localized) would also be helpful in evaluating the impact of this work. For example, on line 79, the Ci centers are described as having an extremely narrow linewidth of 4.2 GHz. This sounds broad compared to a T-center (see Bergeron PRX Quantum 1, 020301 (2020)). If this is extremely narrow for a Ci center, then a comparison to existing values in the literature is appropriate. If not, then in what sense is it extremely narrow?

2- One of my primary criticisms of the manuscript is that much of the work is focused on the Ci center, which is touted as a promising spin-photon interface (line 56), but I find the evidence to support this claim weak. The evidence for the suitability of the Ci for spin-photon purposes is the EPR results shown by Watkins and Brower. These EPR results are insufficient evidence to demonstrate suitability as a spin-photon interface – there is no optical coupling in EPR. Therefore, I find that the claims that the Ci center is a promising spin-photon interface must be better supported for me to consider them credible.

Data analysis, interpretation, and conclusions:

I break up my comments on this into sectioned comments surrounding the figures.

Figure 1

3- There are significant problems with Figure 1. The structures shown as representing the Ci center and the G-center are incorrect. They are also inconsistent with the figure caption. However, the diagram showing the Ci center in Figure 3 a is correct. These must be reconciled. The Ci center shown in Figure 1 is a single substitutional atom (colored blue), but it is described as green and interstitial in the caption. The G-center also needs to be corrected. The description in the text (lines 209-211) is correct but is not reflected in Figure 1. In Figure 1, the pink silicon is drawn as substitutional instead of interstitial.

4- The photoluminescence (PL) inset shown in Figure 1 is difficult to evaluate. Converting the color units into actual counts would be helpful (the excitation power will be necessary to provide in the case).

5- I could not find information about the fluence per pulse from the bottom to the top row. This should be made clear in the text, the figure caption, or the figure itself. The image itself has confusing features. The PL is not constant across the rows despite the fluence remaining the same – is this an artifact of the poor alignment in the imaging system or a real feature of the laser

irradiation? If it is artificial, then I would not support the publication of this PL image. If it is real, then it should be commented on.

6- There is a minor issue with the color scale – a white bar is at the very top of the scale.

7- Additionally, the left-to-right ordering when describing the workflow is reversed in the Figure 1 caption and the image. For example, compare lines 314-315 with the image. The text says the erasure happens on the left, but the erasure happens on the right in the image.

Figure 2

8- It is unclear what the green bar in Fig. 2b represents. Presumably no Ci center as in Figure 2c, but this must be indicated. It also needs to be clarified what these color bands mean, even with their labels. Some text in the figure caption explaining this would be appropriate. My impression is these denote points where no Ci center and no G centers exist in the material. How are these chosen? More elaboration on these criteria and their meaning would be helpful.

Figure 3

DFT is outside of my area of expertise, so I cannot make informed comments on the validity of the approach. However, the thorough study of hydrogen's role in the Ci center and the modified versions of the Ci center structure is fascinating. I believe this is a compelling section of the manuscript.

Figure 4

9- The wavelength scales in Fig. 4a and c are different, but given the comparative claims in Fig4c, I think it is natural to want them on the same wavelength scale. Given that a key claim of Fig4c is that W centers are formed at these high laser fluences, it is necessary to see this range in the PL spectra at lower fluences (like Fig4a). The shown data does not support the claim that W centers appear **only** at higher laser fluences.

General comments

10- Are the errorbars shown in Fig2b+d and Fig 4b+d representative of sample-to-sample deviation or measurement error. I think a comment on this would be useful.

Summary:

The impact of the results presented here is contingent on two main points, which are insufficiently addressed in the current version of the manuscript.

I) How viable is the Ci center as a spin-photon interface?

II) How do these center generation methods compare to existing methods?

Without better clarification on these two points, it is challenging to say the article will likely be impactful.

Additionally and importantly, the mistakes in Figure 1 are egregious. These must be fixed prior to any publication.

REVIEWER COMMENTS

Reviewer #1 (Remarks to the Author):

This work reports local writing and erasing of silicon emitters in the telecom spectral windows using fs laser pulses. The authors demonstrate the controllable formation of G and C_i color centers depending on the forming gas annealing during ion implantation and laser pulse fluence. The authors also explain the bright emission of the C_i centers due to the presence of hydrogen. The approach demonstrated in this manuscript could be potentially used for large-scale integration of quantum emitters in silicon. I have the following comments and questions to the manuscript.

1. Though the authors propose to use their approach for scalable quantum networks, single photon emission is not demonstrated in the manuscript. The C_i centers in Fig. 2a are observed in the pre-processed SOI without fs laser. With fs laser, the C_i centers are erased, but there no strategy described, how can it be used for the controllable creation of single C_i centers.

We acknowledge the valuable comments provided by the esteemed reviewer. We now present data on single center properties for C_i and G centers (new sec. 3 and suppl. info. 6). We show that single centers can be formed reliably with fs laser pulses of relatively low intensity, ~10 mJ/cm² in forming gas treated SOI. This low laser fluence is well below the damage threshold of silicon (~100 to 300 mJ/cm² for fs laser pulses used in our experiments). In situ PL feedback has been combined earlier with laser processing for the controlled, deterministic formation of single quantum emitters in diamond (<https://doi.org/10.1364/OPTICA.6.000662>), and a similar approach could now also be adapted in silicon. We have not conducted these experiments, but we have identified a regime of laser pulse intensities where single center formation on demand can be conducted. We have added a corresponding statement in the discussion section.

2. The authors show that the fs irradiation with laser pulses of an implanted and annealed SOI sample leads to the formation of G centers. But single photon emission is not demonstrated. The authors should describe the strategy how single G centers can be isolated using their approach. In addition to that, the spectral width of the zero-phonon line of the G centers is larger than that of the G centers created by well-established methods. In a similar approach [arXiv:2304.03551], the spectral quality seems to be better. Can the authors comment on this?

As for C_i-centers, we have now also added results on single G center properties following single center writing with fs laser pulses (new sec. 3). We observe linewidths of ~1 nm, which is comparable to the ensemble linewidths observed in the above reference (arXiv:2304.03551) and it is broader than linewidths in several earlier ion beam-based G center formation experiments (<https://doi.org/10.1103/PhysRevApplied.20.014058>). We also present results on the ensemble

linewidths scaling with increasing laser intensity (see suppl. info. 8) in the regime of much higher intensities above the melting threshold of silicon. We believe that an exciting finding of our study is the formation of (single) G and C_i centers at relatively low laser intensities. The linewidths of (single) C_i centers are found to be limited by the spectrometer resolution of ~ 0.03 nm (at 1200 g/mm). The reason for this broader linewidth for the G center in our present experiments might lie in the presence of hydrogen in our SOI samples after forming gas annealing and the role of hydrogen in passivating G centers, but we do not have a quantitative explanation for it at this time. We have also done additional DFT calculations to compare linewidth broadening under strain and stress for the C_i center and compare it with the G center. Relatively narrow linewidth distribution was observed for the C_i center in comparison to the G center (see suppl. Info. 5.3).

3. I find many self-citations of the authors, but no citations of some earlier achievements of other groups that seem to be relevant to their work

- Opt. Express 28, 26111 (2020): isolation of single G centers in silicon
- New J. Phys. 23, 103008 (2021): co-implantation of carbon and hydrogen for the formation of T center in silicon
- Nat. Commun. 14, 2380 (2023): spectrally programmable quantum emitters in silicon
- Nat. Commun. 14, 361 (2023): theory of the formation of carbon-related color centers in silicon

Thank you for the feedback on citations. We've included suggested references and addressed concerns raised by the reviewer. We believe that self-citations are integral to showcasing the evolution and context of our research, considering the diverse nature of the current manuscript. The suggested citations have now been added to the revised manuscript.

4. The use of the words “quantum emitter” in the title and in the main text regarding the experimental results is misleading. The C_i centers emit photons as any ensemble of color centers, but where is “quantum emission” there? In addition to that, “programmable” means that the C_i and G centers center can be activated and erased on demand, independent on the previous state. Particularly, the reactivation after erasing should be demonstrated and the number of such cycles should be discussed. Otherwise, the use of word “programmable” in the title is also misleading. Perhaps, “reconfigurable” or similar would be better.

We agree with the reviewer's suggestion to reserve the term "quantum" specifically for the single-center G and C_i centers. While ensemble emission is also a quantum phenomenon, we acknowledge that the term has been mainly used for isolated centers in the literature. However, with our new demonstration of the formation of isolated/single G and C_i centers post fs laser annealing (sec. 3), we believe the term "quantum" is now apt in our manuscript.

We apologize for any confusion caused by the term "programmable". To support the correct selection of the word "programmability", we would like to bring the reviewer's attention towards the extensive additional exploration of wide range of forming gas annealing and fs laser fluences that we have conducted. We show that we can program, or select the presence of specific quantum emitters in SOI (shown for single G and C_i centers, as well as ensemble H, T, and W centers) and that we can further program center population through writing and erasing of selective quantum emitters. We agree that more follow-up work can show progressively increasing levels of control of single center arrays, e.g. with incorporation of in situ PL feedback, and this is outside the scope of the present study. We do believe though that the term "programmable" accurately reflects our findings of being able to select, control, write, and erase specific quantum emitters in the family of carbon-based quantum emitters in silicon (see suppl. sec. 7 for details).

5. The authors explain the high brightness of the C_i center due to the distortion by the hydrogen atom. The transition dipole moment is calculated for different configurations. To understand the feasibility of the isolation of single C_i centers, it is necessary to compare it with other centers, for instance, G and/or T centers.

Having included data on single C_i and G center properties, we can compare the relative brightness of C_i and G center in our experiments, where we found similar brightnesses of the G and the C_i centers. In earlier DFT calculations Ivanov et al., (<https://arxiv.org/abs/2303.16283>) presented results on transition dipole moments for both G and T centers, finding them to be also comparable. We can thus state that all three centers, C_i , G and T show similar brightnesses (a statement is added in lines 278-283 of the main manuscript).

We note that the relatively lower brightness of the T-center reported in the literature compared to the G-center seen in experiments does not seem consistent with the theoretically computed transition dipole moments (TDMs), which are comparable for the two centers. The reason stems from the nature of the transition in the T-center which involves a localized electron and delocalized hole state [Phys. Rev. Materials 6, L053201, Phys. Rev. B 106, 134107]. This highly delocalized state leads to a finite size error due to the limited size of the computational supercell and additionally leads to an error in the calculation of the TDM. For the G-center and C_i -center, the optical transitions occur between localized states, so our method is suitable to use for these defects.

6. If the presence of hydrogen for high brightness of the C_i centers is important, the dependence on the hydrogen concentration should be provided or at least discussed. Is there a difference if hydrogen is implanted instead of using it as forming gas during the annealing process? What is the dependence on the H_2/N_2 ratio in the forming gas?

We thank the reviewer for expressing this important concern. It's been reported several times in the literature that the hydrogen implantation (proton irradiation) leads to direct formation of G centers and no C_i centers were formed with the same method (to the best of our knowledge). We believe that this is likely due to lattice damage associated with proton implantation. Ion implantation offers more precise fluence control than forming gas annealing (or hydrogen plasma treatment) for the introduction of hydrogen into silicon. However, the associated damage impacts the formation kinetics of selected defects. G-centers form readily after proton irradiation with and without additional thermal annealing, C_i centers do not (<https://doi.org/10.1103/PhysRevB.97.035303>, <https://doi.org/10.1103/PhysRevApplied.20.014058>, <https://doi.org/10.1364/OE.482311>). We have included additional data on G, T, H, and C_i center formation for a series of forming gas annealing conditions in Suppl. Info. 1 and 7.

Summarizing, the authors present a potentially interesting approach for the fabrication of telecom emitters in silicon and control their brightness. On the other hand, the “programmable” and “quantum emitter” are not really demonstrated in the manuscript and the authors oversell their results. Though I think that all minor issues can be resolved in the revised version, the experimental verification of quantum emission requires new experiments. Because this missing piece is the most important part, I do not recommend publication in Nature Communications in the present form.

We appreciate and acknowledge all the comments provided by the reviewer. In this revised manuscript, which includes data from additional measurements, we now show single-center-level control and verify the quantum emitter properties of isolated centers (single photon emission,...). We show that by selecting parameters for forming gas annealing of SOI we can populate specific centers, passivating G centers, and forming C_i and now also H, and T centers. We can then act on these center populations with fs laser pulses, writing and erasing centers, and forming single centers at relatively low laser fluences. We show that the combination of forming gas annealing (introducing new hydrogen bonding configurations) and fs laser pulses of varying intensity allows us to control or program the presence of selected centers. We identify a repeatable process for writing and erasing of these emitters well below the damage threshold of silicon. Hence we believe that the term “programmable” is useful to describe this approach.

Reviewer #2 (Remarks to the Author):

In the manuscripts, the authors demonstrated two main results in the experiments:

1. Local writing and erasing of G centers and C_i centers using fs laser pulses in pre-processed SOI sample
2. Writing and erasing of G centers with direct fs laser pulses in as-received SOI

The main experimental data is the PL spectra from emitters induced by different laser fluences. I think there are some loose arguments in the manuscript. Please find comments below:

1. The title of the manuscript is “Programmable quantum emitter formation in silicon”. However, there is no evidence that the defects induced by fs pulse have quantum nature. To prove this claim, HBT experiments are necessary to verify whether the defects possess antibunching properties. Therefore, at least at current stage, the author's discussion in terms of quantum emitters is inappropriate. In addition, the authors do not fully demonstrate the programmable features. The authors need to explicitly give the repeatability, spatial positioning accuracy, and wavelength stability of the color centers prepared by this method. At the same time, I suggest that the authors give the intensity uniformity of the arrays of color centers by fluorescence mapping (a better demonstration than Fig.1).

We value the constructive feedback from the reviewer and acknowledge the recommendation made regarding the use of the term "quantum". As outlined above in the response to reviewer 1, we have now conducted HBT experiments, demonstrating antibunching properties for single G and C_i centers post-fs laser annealing at low fluences. These results are now incorporated into Fig. 3 and sec. 3 in the revised version (may also refer to response 1, reviewer 1).

We apologize for any confusion caused by the term "programmable". To support the correct selection of the word “programmability”, we would like to bring the reviewer’s attention towards the extensive additional exploration of a wide range for fs laser fluence and of forming gas-based thermal annealing parameters which enable us to select specific windows for erasing and writing of selective quantum emitters. We hence feel that the concept of programmability is useful to describe our approach (please see also our response to reviewer 1 above and suppl. sec. 7).

Fig. 1 inset is now updated and shows single G and C_i center formation after fs pulse irradiation around 12 mJ/cm².

2. Ref 4 reported the creation of W and G-centers by femtosecond laser annealing for the first time on SOI substrates, taking both C-doped and pristine SOI wafers. They can selectively erase the G centers while improving the quality of the W centers’ emissions. The author should be more explicit about the innovations and improvements of this manuscript compared to the previous works.

We appreciate the reviewer for emphasizing the need for explicitness regarding innovations and improvements compared to other recent work (Ref 4).

Our work shows that G and C_i centers can be formed and erased at relatively low laser fluences of about 10 mJ/cm², at intensities orders of magnitude lower than in earlier work and much below the damage threshold of silicon. We believe that this enables repeated writing and erasing of desired quantum emitters in quantum devices without damage to the silicon lattice. We show

this by the use of hydrogen, a well-known ingredient in defect engineering, together with low-intensity fs laser pulses that form single C_i and G centers depending on the laser intensity.

Our findings highlight a sweet spot in the laser fluence range much below the Si damage threshold. At this level, we achieve localized passivation and de-passivation of selected quantum emitters at the single-center level. Contrastingly, higher laser fluences have been reported to lead to the formation of ensembles of W and G centers (see also suppl. sec. 8 and 9).

Furthermore, our contribution extends to introducing hydrogen-based annealing, playing a pivotal role in the activation and passivation of selective emitters. This facet allows exploration of different color center candidates without being overshadowed by intense W and G center emissions common in traditional processes involving C implantation and thermal annealing under inert gases.

Our work delves into the formation of a diverse set of color centers in silicon, such as G, C_i , T, and H centers, through varying forming gas annealing conditions. (See suppl. Info. 7).

3. The most significant data in the manuscript is the variation of the spectra of laser-induced sites with the laser power. As for the fabrication method of quantum emitters in silicon, more optical properties characterization of quantum emitters are needed for a more comprehensive statement of the laser-created color centers in silicon. For example, the long-term stability of the color centers under the light excitation. The current measurements made are too basic to substantiate the claims of significance.

We agree with the reviewer that the variation of the spectra along with the evolution in the optical lifetime and linewidths were among the most significant data in the previous version of the manuscript however another important aspect of the manuscript is also to understand the role of hydrogen in the formation of selective quantum emitters by passivating the most commonly formed G center from conventional approaches. The role of hydrogen is of utmost importance to locally passivate and de-passivate G and C_i centers at low fs laser fluences. Now in the revised version, we have included additional measurements of optical properties such as lifetime, and linewidth, dipole orientation, along with HBT measurements of single G and C_i centers. We have measured PL from a single C_i center for 30 min, observing stable emission, and have included this data point now in supplementary sec. 6.3.

4. Line 97-192 in the main text: Considering the bandgap of silicon, the main absorption mechanism of femtosecond lasers at 800 nm should not be multiphoton absorption. At the same time, the discussion of the physical mechanisms in this section is too general to be convincing. The authors should give a specific theoretical analysis and simulation of the interaction mechanism in order to relate it to the high-quality center generation. The same problem occurs in the discussion of linewidths, where the authors do not give a convincing reason for the physical origin of such a narrow linewidth.

We appreciate the critical remark made by the reviewer and agree that indeed the single photon absorption will be the dominating process, we have now done additional phenomenological calculations based on the laser fluence regime to explain the underlying formation mechanism. A detailed explanation is now added in the manuscript sec. 3 (line 108-128).

For the discussion on linewidths, we have now done DFT calculations to compare the linewidth broadening of the C_i center under strain and stress with the G center. A relatively narrow broadening was found for the C_i center (See suppl. Info. 5.3 for details).

5. Does the transition from single color center to multiple color centers occur as the laser energy increases?

Indeed, at higher laser fluences, the color centers are mostly dominated by ensemble W and G centers. The single center level is achieved with much lower fluences, many folds below the damage threshold of Si. This is also in line with the Keldysh parameter estimation where the K value of above 1.5 refers to the photoionization process via single photon absorption which is also referred to as the main reason behind writing single color centers in other host materials.

Reviewer #3 (Remarks to the Author):

Jhuria et al. report on generating interstitial carbon centers (C_i) and G-centers in silicon using femtosecond (fs) pulsed lasers. They also report on a hydrogen-based annealing recipe that enhances the C_i centers and suppresses the G-centers. Additionally, they present density functional theory calculations that analyze the C_i center in the presence of hydrogen. Significance:

As noted by the authors, pulsed laser annealing for writing color centers has been widely studied in many wide-bandgap materials already (for example, see Castelletto et al, Nanomaterials (Basel). 2021 Jan; 11(1): 72). Nevertheless, examining the role of fs laser irradiation in silicon specifically and the possible centers available for writing is a valuable exercise as silicon-based color centers occupy an increasingly prominent role in the color center community.

1- It would be useful to have a short comparison of figures of merit to the author's previous published work generating G-centers using pulsed ions – i.e., how do the centers compare? Additional context for the quality of the color centers created using this technique compared to other methods (non-localized) would also help evaluate the impact of this work. For example, on line 79, the C_i centers are described as having an extremely narrow linewidth of 4.2 GHz. This sounds broad compared to a T-center (see Bergeron PRX Quantum 1, 020301 (2020)). If this is extremely narrow for a C_i center, then a comparison to existing values in the literature is appropriate. If not, then in what sense is it extremely narrow?

We thank the reviewer for this comment. We observe C_i centers with a linewidth of 0.03 nm (limited by our spectrometer, see suppl. sec. 3), which is narrower than that observed for single

G centers, and both are much broader than T centers (in 28-Si). We have thus removed the phrase “extremely narrow”.

C_i centers formed here were not observed at all in our earlier work with (pulsed) ion beams [<https://www.nature.com/articles/s43246-023-00349-4>, <https://doi.org/10.1103/PhysRevApplied.20.014058>]. G-centers in ensembles observed here are slightly broader than those we had formed e. g. by proton irradiation (without thermal annealing). In our present work, we show trends in ensemble center linewidths vs. fs laser intensity, which can be compared also to trends from ion beam-based processing as a function of ion fluence (and associated damage and buildup of disorder, see suppl. sec. 4, 8).

2- One of my primary criticisms of the manuscript is that much of the work is focused on the C_i center, which is touted as a promising spin-photon interface (line 56), but I find the evidence to support this claim weak. The evidence for the suitability of the C_i for spin-photon purposes is the EPR results shown by Watkins and Brower. These EPR results are insufficient evidence to demonstrate suitability as a spin-photon interface – there is no optical coupling in EPR. Therefore, I find that the claims that the C_i center is a promising spin-photon interface must be better supported for me to consider them credible.

Data analysis, interpretation, and conclusions:

We appreciate this comment and concern that the promise of the C_i center as a spin-photon qubit candidate is not well enough established at this point. Yes, the earlier EPR work is promising, but this alone is insufficient for a spin-photon qubit. We, now show access to single quantum emitters, as well as DFT calculations that include spin and properties of H-modified C_i centers. We believe that together this makes the C_i -H center a promising candidate. We also fully agree that more work is needed to quantify and map out the level structure of the C_i -H center (as it has been done and is subject to ongoing work for other very promising centers, such as the T center, and the earlier work on NV centers in diamond etc.).

In addition, our finding of the role of hydrogen to passivate and activate selected centers, together with the ability to act on these centers with relatively low-intensity laser pulses, also opens the door to explore such candidates by selectively passivating and de-passivating different quantum emitters at the desired locations. While the concern raised by the reviewer is important regarding the spin-photon coupling efficiency, it is a separate demanding project and beyond the scope of the current work.

I break up my comments on this into sectioned comments surrounding the figures.

Figure 1

3- There are significant problems with Figure 1. The structures shown as representing the C_i center and the G-center are incorrect. They are also inconsistent with the figure caption.

However, the diagram showing the C_i center in Figure 3 a is correct. These must be reconciled. The C_i center shown in Figure 1 is a single substitutional atom (colored blue), but it is described as green and interstitial in the caption. The G-center also needs to be corrected. The description in the text (lines 209-211) is correct but is not reflected in Figure 1. In Figure 1, the pink silicon is drawn as substitutional instead of interstitial.

4- The photoluminescence (PL) inset shown in Figure 1 is difficult to evaluate. Converting the color units into actual counts would be helpful (the excitation power will be necessary to provide in the case).

5- I could not find information about the fluence per pulse from the bottom to the top row. This should be made clear in the text, the figure caption, or the figure itself. The image itself has confusing features. The PL is not constant across the rows despite the fluence remaining the same – is this an artifact of the poor alignment in the imaging system or a real feature of the laser irradiation? If it is artificial, then I would not support the publication of this PL image. If it is real, then it should be commented on.

6- There is a minor issue with the color scale – a white bar is at the very top of the scale.

7- Additionally, the left-to-right ordering when describing the workflow is reversed in the Figure 1 caption and the image. For example, compare lines 314-315 with the image. The text says the erasure happens on the left, but the erasure happens on the right in the image.
Figure 2

We deeply thank the reviewer for pointing out the mistakes in Fig. 1. We have taken comments 3-7 into account and updated Fig. 1. The hyperspectral PL image In Fig. 1 is now updated with a higher quality scan showing spots with a distribution of (single) C_i centers from fs irradiation at a laser fluence of 12 mJ/cm^2 . Schematics representation of the G, and C_i centers are also updated and corrected.

8- It is unclear what the green bar in Fig. 2b represents. Presumably no C_i center as in Figure 2c, but this must be indicated. It also needs to be clarified what these color bands mean, even with their labels. Some text in the figure caption explaining this would be appropriate. My impression is these denote points where no C_i center and no G centers exist in the material. How are these chosen? More elaboration on these criteria and their meaning would be helpful.

Figure 2 is now updated with corrections regarding the color bands and a proper explanation on the same is now added in the caption of the figure as well as on the figure is now included.

Figure 3

DFT is outside of my area of expertise, so I cannot make informed comments on the validity of the approach. However, the thorough study of hydrogen's role in the Ci center and the modified versions of the Ci center structure is fascinating. I believe this is a compelling section of the manuscript.

We highly appreciate this acknowledgment.

Figure 4

9- The wavelength scales in Fig. 4a and c are different, but given the comparative claims in Fig4c, I think it is natural to want them on the same wavelength scale. Given that a key claim of Fig4c is that W centers are formed at these high laser fluences, it is necessary to see this range in the PL spectra at lower fluences (like Fig4a). The shown data does not support the claim that W centers appear *only* at higher laser fluences.

We appreciate the reviewer for their in-depth analysis of the manuscript. Since we now have a single center level control, we have decided to move the section related to current Fig. 4 to the suppl. Info. 9 with full scale from 1200-1300 nm showing both W and G center range. The only reason to not include it in the figure was for better visibility in the previous version.

General comments

10- Are the errorbars shown in Fig2b+d and Fig 4b+d representative of sample-to-sample deviation or measurement error. I think a comment on this would be useful.

Error bars shown throughout the figures are statistical and include experimental error including sample-to-sample deviation and also the fluctuation in the laser pulse power/fluence (where the fluctuation from the fs laser source is the major factor in the current situation that can be readily controlled by using a more stable laser source. A comment is now added in the figure caption.

Summary:

The impact of the results presented here is contingent on two main points, which are insufficiently addressed in the current version of the manuscript.

- I) How viable is the Ci center as a spin-photon interface?
- II) How do these center generation methods compare to existing methods?

Without better clarification on these two points, it is challenging to say the article will likely be impactful.

Additionally and importantly, the mistakes in Figure 1 are egregious. These must be fixed prior to any publication.

We sincerely appreciate the reviewer's constructive feedback, which is instrumental in enhancing the manuscript.

ad 1) We present data on single C_i centers after forming gas annealing of SOI, and establish a methodology for their reliable formation with relatively low-intensity laser pulses. The linewidths we observed of 0.03 nm for these single centers are limited by our spectrometer resolution. We present DFT calculations that include spin and optical properties of C_i centers. While earlier work has established spin properties in EPR measurements, detailed studies of the level structure of C_i centers are not available yet, and these will be required to qualify C_i centers as potentially viable spin-photon interfaces. Yet, we argue that the results we present here significantly advance the field by demonstrating the role of hydrogen in quantum emitter passivation and activation together with low-intensity laser pulses at the single center level. We believe that this novel approach enables controlled writing and erasing of single centers and the programming of center configurations for quantum applications.

ad II) Having gained experience with (single) quantum emitter formation using proton and ion beams as well as now fs laser pulses, we can state that local programming of center populations by selective writing and erasing of desired centers is uniquely enabled by the fs laser approach in hydrogen treated SOI. (Focused) ion and proton beams can locally form quantum emitters, and can also erase them at progressively higher ion fluences, but both are associated with significant lattice damage, often necessitating the use of thermal annealing (during which new centers can form). The low intensity fs laser process we have pioneered allows us to form single centers reliably without further annealing.

In contrast to protons, ion beams and high power laser pulses, we show that fs laser pulses at relatively low intensity can “gently” form and erase quantum emitters by acting on the hydrogen, well below the threshold intensity for the onset of melting and lattice damage. We show this for the C_i center, and we expect this approach to enable the discovery, study, and then integration of other potentially promising quantum emitters.

REVIEWERS' COMMENTS

Reviewer #1 (Remarks to the Author):

In the revised manuscript, the authors have addressed all the issues raised by the reviewers. In particular, I would like to emphasise the new experiments demonstrating single photon emission, which were missing in the first version of the manuscript. Therefore, I recommend publication in Nature Communications in its present form.

Reviewer #2 (Remarks to the Author):

The authors have addressed my concerns properly and the manuscript is ready for publication. Furthermore, a recently published study has shown that femtosecond lasers can generate high-quality single-photon sources on hBN with sub-10 nm precision [Laser manufacturing of spatial resolution approaching quantum limit, DOI:10.1038/s41377-023-01354-5]. In the discussion or conclusion part, the authors may consider exploring the potential of realizing quantum light sources in silicon with even higher spatial precision in the future, building upon the findings presented in this article.

Reviewer #3 (Remarks to the Author):

Jhuria et al. have addressed my concerns, and I can now fully recommend publication. Additionally, I think the g2 measurements added in Figure 3 (and the changes to Fig. 3 overall) considerably strengthen the manuscript.

1 **REVIEWER COMMENTS**

2

3 Reviewer #1(Remarks to the Authors)

4 In the revised manuscript, the authors have addressed all the issues raised by the reviewers. In
5 particular, I would like to emphasise the new experiments demonstrating single photon emission,
6 which were missing in the first version of the manuscript. Therefore, I recommend publication in
7 Nature Communications in its present form.

8 We highly appreciate the remarks and recommendation for publication made by the esteemed
9 reviewer, indeed the formation of these quantum emitters at single centers level is the major
10 highlight in the article.

11 Reviewer #2

12 The authors have addressed my concerns properly and the manuscript is ready for publication.
13 Furthermore, a recently published study has shown that femtosecond lasers can generate high-
14 quality single-photon sources on hBN with sub-10 nm precision [Laser manufacturing of spatial
15 resolution approaching quantum limit,DOI:10.1038/s41377-023-01354-5]. In the discussion or
16 conclusion part, the authors may consider exploring the potential of realizing quantum light
17 sources in silicon with even higher spatial precision in the future, building upon the findings
18 presented in this article.

19 We appreciate the remarks and recommendation for publication made by the reviewer. We
20 appreciate the mention of the work on high resolution laser writing and have now included this
21 reference in the conclusion section of the main manuscript (line 275, new reference 50, Wang,
22 X. J., Fang, H. H., Li, Z. Z., Wang, D. & Sun, H. B. Laser manufacturing of spatial resolution
23 approaching quantum limit. *Light Sci. Appl.* **13**, (2024).).

24 Reviewer #3

25 Jhuria et al. have addressed my concerns, and I can now fully recommend publication.
26 Additionally, I think the g2 measurements added in Figure 3 (and the changes to Fig. 3 overall)
27 considerably strengthen the manuscript.

28 We appreciate the remarks and recommendation for publication made by the reviewer.